# SAFETY-ADVANCED AUTONOMOUS DRIVING FOR URGENT HAZARDOUS SITUATIONS USING Q-COMPARED SOFT ACTOR-CRITIC

## ABSTRACT

Autonomous vehicles must be capable of safe driving under all conditions to ensure passenger safety. This includes urgent hazardous situations (UHS), such as skidding on slippery roads or tire grip saturation during high-speed driving, which are not only difficult even for expert human drivers but also challenging to develop autonomous driving technologies that surpass human capabilities. Even though the recent advancements in machine learning including imitation learning (IL), reinforcement learning (RL), and hybrid learning (HL) have enabled the safe navigation of autonomous vehicles in various complex scenarios, they have fundamental limitations in UHS. Driving policies trained via IL degrade in novel situations where expert demonstration data is scarce or of poor quality, and RL struggles to develop optimal driving policies in UHS, which have broad state and action spaces and high transition variance. HL techniques combining IL and RL also fall short, as they require nearly optimal demonstration data, which is nearly impossible to obtain in UHS due to the difficulty for human drivers to react appropriately. To address these limitations, we propose a novel HL technique, Q-Compared Soft Actor-Critic (QC-SAC), which effectively utilizes immature demonstration data to develop optimal driving policies and adapt quickly to novel situations in UHS. QC-SAC evaluates the quality of demonstration data based on action value Q to prioritize beneficial data and disregard detrimental ones. Furthermore, QC-SAC improves the performance of the Q-network by leveraging demonstration data and enhances learning by rapidly incorporating new successful experiences from ongoing interactions, enabling fast adaptation to new situations. We test QC-SAC for two extreme UHS scenarios: oversteer control with collision avoidance (OCCA) and time-trial race (TTR). In OCCA, QC-SAC achieves a success rate 2.36 times higher than existing techniques, and in TTR, it reduces lap time by more than 13.6% while completing 300 test runs without a single failure. By proposing an innovative HL technique capable of training superior driving policies with immature demonstration data, we provide a solution for autonomous driving technologies that can handle UHS and introduce the world-first safe-advanced autonomous driving technology capable of controlling a vehicle oversteer safely and avoiding obstacles ahead.

## 1 INTRODUCTION

Driving often involves encountering various hazardous events that usually lead to fatal accidents. For instance, in emergency situations, where a vehicle skids on icy or wet roads or suddenly encounters obstacles, even skilled human drivers find it challenging to react safely. While current autonomous driving technologies show reliable driving performance in everyday road conditions, addressing such urgent hazardous situations (UHS) still remains a big challenge. And it is generally assumed that the autonomous vehicles should transfer the vehicle control to human drivers in emergency situations including the UHS. However, studies indicate that it takes human drivers approximately 6 to 7 seconds, or even as long as 12 to 15 seconds, to perceive the situation and start vehicle control (Kuehn et al., 2017). This implies that requesting a takeover to human drivers is impractical in

UHS that could lead to severe accidents in less than a couple of seconds. Therefore, it is crucial for autonomous driving agents to perceive the UHS and apply appropriate vehicle control immediately.

Recently, with the advancement of machine learning, including imitation learning (IL) and reinforcement learning (RL), it has become possible to navigate the autonomous vehicle safely in various complex driving scenarios (Chib & Singh, 2023; Le Mero et al., 2022; Zhu & Zhao, 2021; Kiran et al., 2021), or sometimes outperform human drivers (Wurman et al., 2022; Kaufmann et al., 2023). Despite these achievements, both IL and RL have fundamental limitations. The action policies trained via IL often face significant performance degradation when they encounter novel situations where expert demonstration data is scarce or when the quality of the demonstration data is poor. On the other hand, RL requires sufficient experience of successful episodes. However, as the task becomes more complex, the state and action spaces become broader, and the transition dynamics have higher variance, the probability of experiencing successful episodes through random actions diminishes. Consequently, developing an optimal action policy using RL becomes exceedingly difficult (Huang et al., 2023; Zhao, 2021). To overcome these shortcomings, recent hybrid learning (HL) techniques combine IL and RL, utilizing expert demonstration data for more efficient and improved policy development (Hester et al., 2018; Rajeswaran et al., 2017; Alakuijala et al., 2021; Tian et al., 2021; Lu et al., 2023; Gao et al., 2018). However, HL techniques have a limitation in that the expert demonstrations must be nearly optimal or, even if noisy, must contain optimal demonstration data (Gao et al., 2018).

Such limitations in IL, RL, and HL become more critical in UHS. Firstly, it is nearly impossible in the case of UHS to collect optimal demonstration data, because human drivers find it difficult to react appropriately and try immature vehicle control. As a result, the driving policy trained via IL or HL techniques could perform poorly. Moreover, in UHS, there is an extensive amount of information to be perceived regarding the surroundings and the vehicle state, and there is a high transition variance due to the nonlinear dynamics. Not only does this make it challenging to develop an optimal driving policy using RL alone, but it also means that agents are unlikely to encounter the same situation multiple times, making it difficult to learn through repeated experiences. Therefore, there is a critical need for methods that enable fast learning from new situations and can rapidly incorporate new successful experiences into the learning process.

To overcome the aforementioned limitations, we propose an innovative HL technique, Q-Compared Soft Actor-Critic (QC-SAC), that can learn optimal policies even from immature demonstration data and adapt quickly to novel situations, which are common in UHS. To demonstrate the superior performance of QC-SAC, we evaluate the proposed technique in two extreme UHS scenarios: oversteer control with collision avoidance (OCCA) and time-trial race (TTR). In OCCA, where even human drivers often fail to control the vehicle and lead to severe accidents, a test vehicle equipped with the driving policy developed by the proposed QC-SAC should successfully avoid obstacles ahead when it suddenly starts to spin (oversteer) on a slippery road. In TTR, where the tires of a vehicle reach their frictional limit and become difficult to control, the driving policy must appropriately and precisely control the vehicle and achieve the fastest lap time. The severity and importance of the two scenarios are discussed in more detail in section 2.

With successful demonstrations in these two scenarios, we prove that the proposed QC-SAC can develop an optimal driving policy for UHS that applies appropriate and prompt action control and safely navigates the vehicle. Note that this study introduces the world-first safe driving technology capable of successful autonomous control of vehicles that should avoid obstacles ahead when it is in oversteer condition.

## 2 PROBLEM STATEMENT

### 2.1 OVERSTEER CONTROL WITH COLLISION AVOIDANCE (OCCA)

Oversteer occurs when the rear wheels of a vehicle slip more than the front wheels, due to the reduced road friction caused by road icing or hydroplaning, tire slippage from sharp steering or pedal manipulation beyond the vehicle's limits, or rear-end collisions, which results in more rotation than the driver's intention. Controlling oversteer requires both appropriate pedal manipulation and adequate counter-steering at the same time (a sophisticated driving technique where the driver steers in the opposite direction of the skid) (Morton, 2006). However, it is very difficult for untrained ordi-

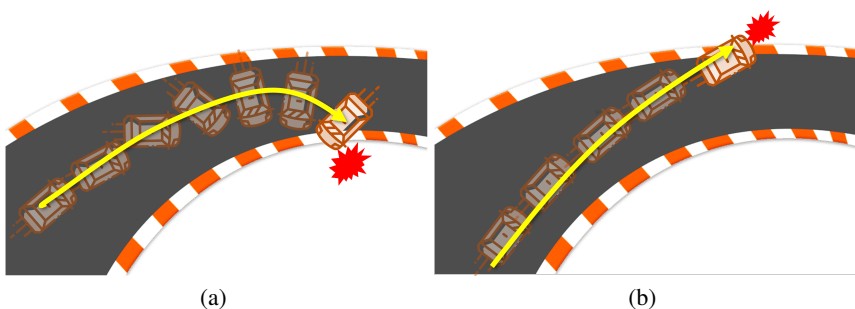

(a)                                                                (b)

Figure 1: Vehicle oversteer and understeer. a, Oversteer: the rear tires lose the grip, and the vehicle rotates more than intended. b, Understeer: the front tires lose the grip, and the vehicle turns less than expected.

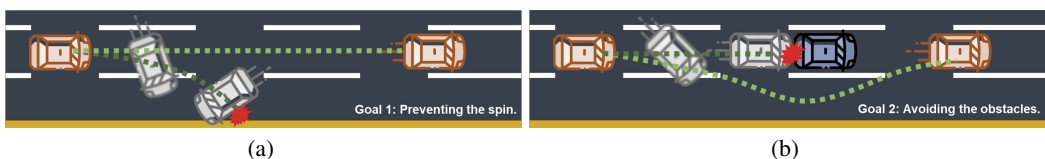

(a)                                                                (b)

Figure 2: Research goals. a, Ego vehicle in brown must control the oversteer in order not to spin. b, It should also avoid the obstacle (i.e., front vehicle) in blue.

nary drivers to use such a specialized driving technique. As a result, oversteer often leads to severe traffic accidents; according to the National Highway Traffic Safety Administration (NHTSA) Fatality Analysis Reporting System (FARS), oversteer accounted for more than 18,852 fatal accidents in the United States from 2011 to 2020, making it the 8th leading cause of fatal accidents.

As autonomous vehicles are not free from oversteer, developing autonomous driving technology capable of handling oversteer is essential for passenger safety and is a challenge. Although several studies have addressed controlling oversteer in autonomous vehicles, most have focused on control techniques rather than path planning (Goh et al., 2018; Velenis et al., 2011; Zhang et al., 2017; Zubov et al., 2018; Zhang et al., 2018; Acosta & Kanarachos, 2018; Cai et al., 2020; Cutler & How, 2016). These studies introduced control techniques for tracking predefined paths (Goh et al., 2018; Velenis et al., 2011; Zhang et al., 2017; Zubov et al., 2018; Acosta & Kanarachos, 2018; Cai et al., 2020; Cutler & How, 2016) or simple paths planned with methods like Rapid Random Tree (RRT) in obstacle-free roads (Zhang et al., 2018). By utilizing such control techniques, an oversteering vehicle can be returned to its original lane, as shown in Figure 2a. However, on real roads with surrounding obstacles (i.e., vehicles), merely controlling an oversteer is not enough to avoid collision with obstacles. When there is an obstacle in the driving lane, as illustrated in Figure 2b, the vehicle must recognize the surrounding environment and find an alternative evasive path that should not only guarantee collision avoidance but also be controllable for the oversteering vehicle to follow. Therefore, this study utilizes an end-to-end approach that maps vehicle state and surrounding environment directly to control by integrating path planning and vehicle control into a single neural network. In other words, the controllability is considered in the path planning simultaneously.

To develop the end-to-end driving policy network capable of oversteer control with collision avoidance, we propose a novel training and testing scenario. The scenario is inspired by the real driver training procedure which utilizes a kick plate: a device that induces oversteer intentionally by moving laterally when the rear wheels of the vehicle pass over it. (see Figure 3a.) The human driver must control the oversteer while avoiding water fountains or virtual obstacles, as illustrated in Figure 3b. For safety reasons, we implement this within a simulator, creating a virtual kick plate to induce oversteer. Obstacles are then randomly placed on the road so that the autonomous vehicle encounters them after the oversteer is induced. A detailed introduction to the scenario is provided in appendix A.2.

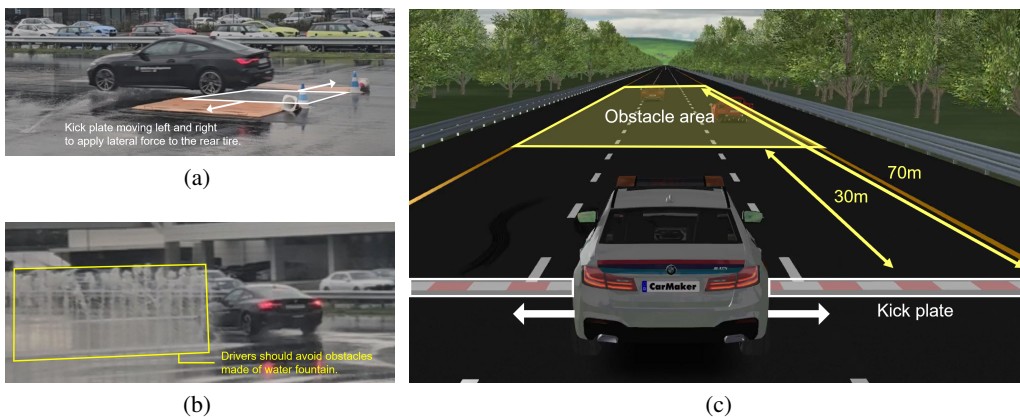

Figure 3: Real-world driver training process and OCCA scenario in a virtual environment. a, Kick plate inducing oversteer, and b, a collision avoidance scenario for driver training at BMW Driving Center, South Korea. c, OCCA scenario developed in IPG CarMaker simulator.

## 2.2 TIME-TRIAL RACE (TTR)

In addition to OCCA scenario, we evaluate the proposed QC-SAC for a different task (i.e., TTR task) in a different environment, which demonstrates the robustness of the proposed QC-SAC to the tasks and environments. TTR aims to complete a certain racetrack in the shortest time possible. In a racing situation, as the vehicle is pushed to the tires' frictional limit for faster driving, vehicle control becomes challenging because of the non-linear dynamic motion of the vehicle (Wurman et al., 2022). Therefore, research on autonomous driving technology capable of fast and safe driving in racing has been active (Wurman et al., 2022; Wischnewski et al., 2022a; Li et al., 2021; Gandhi et al., 2021; Kapania & Gerdes, 2020; Cai et al., 2021; Betz et al., 2022) and demonstrated in various virtual car (Francis et al., 2022), model car (O'Kelly et al., 2020), and real car (Wischnewski et al., 2022b) race competitions. In this study, we evaluate the driving policy developed by the proposed QC-SAC in the challenging TTR scenario. Further details regarding the scenario setup can be found in appendix A.3.

## 3 Q-COMPARED SOFT ACTOR-CRITIC (QC-SAC)

As previously mentioned, it is difficult to develop an optimal driving policy for UHS using existing IL, RL, or HL techniques (refer to Appendix A.1 for a detailed explanation of existing techniques and their limitations). To overcome these technical limitations, we propose a novel HL technique called Q-Compared Soft Actor-Critic (QC-SAC). In this section, we introduce the three core elements of QC-SAC: Q-Compared Objective (QCO), Q-Network from Demonstration (QNfD), and Selective Demonstration Data Update (SDDU). QCO and QNfD contribute to the effective utilization of immature demonstration data to develop an optimal driving policy for UHS, while QNfD and SDDU facilitate the rapid incorporation of new successful experiences into the learning process.

## 3.1 Q-COMPARED OBJECTIVE (QCO)

To effectively utilize immature demonstrations and support RL for optimal policy development with IL, QCO selectively utilizes only the beneficial demonstration data for behavior cloning (BC). It estimates the action value (Q) difference between the actions stored in the demonstration data and the actions generated by the driving policy being developed. Then, it prioritizes the demonstration data that have higher Q values than the driving policy being developed, where the prioritization is implemented by weighting the demonstration data in proportion to the quality difference. QCO thus resolves a critical problem in the conventional HL techniques; the action policy becomes suboptimal when immature demonstration data is provided.

QCO can be expressed as:

$$J_\pi(\phi) = J_{SAC}(\phi) + J_{BC}(\phi), \tag{1}$$

where the objective of the Soft Actor-Critic (Haarnoja et al., 2019) RL is

$$J_{SAC}(\phi) = -\mathbb{E}_{(s,a)\sim\rho_{\pi_\phi}} \left[ Q(s,a) + \alpha\mathcal{H}(\pi_\phi(\cdot|s)) \right], \tag{2}$$

and objective of BC is

$$J_{BC}(\phi) = \mathbb{E}_{(s_d,a_d)\sim\mathcal{D}} \left[ C(s_d, a_d) \cdot \mathcal{L}_1(a \sim \pi_\phi(s_d), a_d) \right]. \tag{3}$$

In (3), $\mathcal{L}_1$ represents the $L_1$ loss, and $C(s_d, a_d)$ is the Q-compared weighting factor for each demonstration data, which can be calculated as below.

$$C(s_d, a_d) = \max \left( Q^-(s_d, a_d) - Q(s_d, a \sim \pi_\phi(s_d)), 0 \right), \tag{4}$$

where $Q^-$ is the Q-target value.

The goal is to determine the parameters $\phi$ of the $\pi$ network that minimize the cost function $J_\pi(\phi)$, which means that we determine $\phi$ that minimizes both the RL objective $J_{SAC}(\phi)$ (2) and the BC objective $J_{BC}(\phi)$ (3). Note that $J_{SAC}(\phi)$ (2) is the conventional SAC objective function, while $J_{BC}(\phi)$ (3) uses $C(s_d, a_d)$ for the Q value comparison. As shown in (4), $C(s_d, a_d)$ uses the action value function Q to evaluate which has the higher Q value between the given demonstration action $a_d$ and the action $a \sim \pi_\phi(s_d)$ of the RL action policy $\pi_\phi$ in the same state $s_d$. By multiplying the BC loss by the difference in two Q values as a weight, the more $a_d$ has a higher Q value than $a \sim \pi_\phi(s_d)$, the more $a_d$ is considered important and is given with higher priority. Additionally, by using the max function in (4), if $a_d$ has a lower Q value than $a \sim \pi_\phi(s_d)$, $a_d$ is considered to deteriorate the training and discarded from the training by setting the weight $C(s_d, a_d)$ to 0.

Additionally, to improve the numerical stability, $L_1$ loss is used for the BC loss in (3). As shown in (11), log probability can be used for BC when demonstration data is optimal. However, when $a_d$ significantly differs from the distribution of $\pi_\phi(s_d)$ because of the immature demonstrations, $\pi_\phi(a_d|s_d)$ approaches to 0 and the log probability diverges. In contrast, $L_1$ loss is limited to the maximum value of 2 because of the action space constrained between -1 and 1, so that the risk of divergence is prevented. Since QC-SAC allows immature demonstrations, $L_1$ loss is used to ensure numerical stability even when immature demonstration data very different from $\pi_\phi(s_d)$ is given. In this manner, the QCO of QC-SAC is specialized for effective utilization of immature demonstrations.

### 3.2 Q-Network from Demonstration (QNfD)

Since Q values are used as the metric for evaluating $a_d$ and $a \sim \pi_\phi(s_d)$ in (4), the Q-network must be well-trained to accurately estimate $C(s_d, a_d)$, thus enabling QCO to achieve high performance. Particularly in UHS situations, where agents frequently encounter novel situations and have difficulties experiencing successful episodes due to broad state and action spaces and high transition variance, it is crucial to obtain as much data as possible to train the Q-network effectively. To achieve this, we propose QNfD method that utilizes demonstration data to enhance the Q-network training.

The concept of utilizing demonstration data for Q-network training is recently proposed, in the HL technique employing IL-based pre-training followed by fine-tuning with RL (Wang et al., 2023). The study shows that when the RL is based on actor-critic, both networks for the actor and the critic need to be pre-trained. In our work, to fit the QC-SAC structure, which combines IL and RL into a single objective function without pre-training, we propose a method that combines two batches for the Q-network update. As expressed in lines 23~25 of Algorithm 1 in appendix A.4, the Q-network is updated using the union $\mathcal{B}$ of the batch $\mathcal{B}_{RL}$ sampled from the replay buffer $\mathcal{D}_{RL}$ collected through interaction with the environment and the batch $\mathcal{B}_{BC}$ sampled from the demonstration dataset $\mathcal{D}$. For this purpose, not only the state and action but also the reward and next state are recorded when collecting the demonstration data. The Q-network is updated using the following conventional objective function used in RL:

$$J_Q(\theta_i) = \mathbb{E}_{(s,a,r,s')\sim\pi}\left[\left(Q_{\theta_i}(s,a) - \hat{Q}(r,s')\right)^2\right], \tag{5}$$

where $Q_(\theta_i)$ is parameterized by $\theta_i$, $i \sim \{1,2\}$ is the index of two Q-networks for double Q-learning (Hasselt, 2010), and

$$\hat{Q}(r,s') = r + \gamma\mathbb{E}_{a'\sim\pi}\left[Q_{\theta_i}^-(s',a') - \alpha\log\pi_\phi(a'|s')\right]. \tag{6}$$

Enhancing the performance of the Q-network through QNfD can significantly improve the performance of QC-SAC, i.e., it can produce more accurate Q value estimates for $C(s_d, a_d)$ and $J_{SAC}(\phi)$ (2) in QCO.

### 3.3 Selective Demonstration Data Update (SDDU)

In urgent hazardous situations (UHS), where the state and action spaces are broad and the transition variance is high, agents often encounter novel situations and find it difficult to learn by experiencing the same situations multiple times. To address this challenge, we propose the SDDU method, which enables rapid learning from new situations and the swift incorporation of new successful experiences into the learning process. SDDU selects successful episodes from interactions with the environment during the training process and uses aggregated data as the demonstration data for training. Before starting the training process, the average episode reward $\bar{r}_{epi}$ of the demonstration data is recorded, which can be calculated as:

$$\bar{r}_{epi} = \frac{1}{|\mathcal{D}|}\sum_{i=1}^{|\mathcal{D}|}\sum_{j=1}^{|\mathcal{D}[i]|} r_{i,j}, \tag{7}$$

where $|\mathcal{D}|$ is the size of demonstration dataset $\mathcal{D}$, $|\mathcal{D}[i]|$ is the number of steps in the $i^{\text{th}}$ episode in $\mathcal{D}$, and $r_{i,j}$ is the reward of the $j^{\text{th}}$ step in the $i^{\text{th}}$ episode.

Subsequently, while the agent interacts with the environment for training, at the end of each episode, the episode reward $r_{epi}$ (i.e., the sum of reward received in that episode) is compared to $\bar{r}_{epi}$ of the dataset. If the reward $r_{epi}$ of the new episode is higher than $\bar{r}_{epi}$, the episode is added to the dataset, and $\bar{r}_{epi}$ is updated again using (7). This iterative process is shown in lines 13~17 of Algorithm 1. By employing SDDU, the agent effectively expands its knowledge base with higher-quality data, enhancing the learning stability and performance of the driving policy. SDDU not only alleviates the data scarcity problem in IL by increasing the size and quality of the dataset, but also contributes to the effective minimization of QCO. After the iterative process of SDDU, we obtain the updated dataset $\mathcal{D}'$, which satisfies $\mathbb{E}_{(s,a)\sim\mathcal{D}'}[r(s,a)] > \mathbb{E}_{(s,a)\sim\mathcal{D}}[r(s,a)]$. Then, from $Q(s,a) = \mathbb{E}\left[\sum_{k=0}^{\infty}\gamma^k r_{t+k}|s_t = s, a_t = a\right]$, it follows that $\mathbb{E}_{(s,a)\sim\mathcal{D}'}[Q(s,a)] > \mathbb{E}_{(s,a)\sim\mathcal{D}}[Q(s,a)]$. Therefore, from (4), $\mathbb{E}_{(s,a)\sim\mathcal{D}'}[C(s,a)] > \mathbb{E}_{(s,a)\sim\mathcal{D}}[C(s,a)]$, and thus $J_{BC}(\phi)$ is considered more importantly, promoting active learning from expert behavior. Additionally, training $\pi_\phi$ by minimizing $J_{BC}(\phi)$ with $\mathcal{D}'$ leads to higher $\mathbb{E}_{(s,a)\sim\rho_{\pi_\phi}}[Q(s,a)]$, contributing to the minimization of $J_{SAC}(\phi)$ as well.

In a summary, the proposed QC-SAC technique consists of three key elements: QCO, QNfD, and SDDU. The overall structure of QC-SAC can be found in Algorithm 1. We use (5) to update the Q-network and (1) to update the $\pi$ network. The temperature $\alpha$ is updated using the same formula of the second version of SAC (Haarnoja et al., 2019).

$$J(\alpha, \bar{\mathcal{H}}) = \mathbb{E}_{a\sim\pi}\left[-\alpha\log\pi(a|s) - \alpha\bar{\mathcal{H}}\right] \tag{8}$$

### 3.4 Focused Experience Replay (FER)

To improve training quality, we employ Focused Experience Replay (FER) (Kong et al., 2021). FER addresses the data imbalance problem in conventional random sampling, where older data in

the replay buffer is more likely to be sampled than recent data. By using a half-normal distribution for sampling, FER prioritizes recent data, enhancing training speed and stability.

# 4 RESULTS

To evaluate the performance of QC-SAC, we compare the performance of driving policy developed with QC-SAC to those of three other representative conventional training techniques: Behavior Cloning (BC), Soft Actor-Critic (SAC) (Haarnoja et al., 2019), and Behavior Cloned Soft Actor-Critic (BC-SAC) (Lu et al., 2023) that are representative and widely used IL, RL, and HL techniques, respectively. SAC is the most widely used RL technique for its strong performance, while BC-SAC is a representative HL technique that combines the objectives of BC and SAC. More detailed descriptions of these existing techniques are provided in section A.1. In this section, we demonstrate the superiority of QC-SAC by comparing the performance of driving policies in the OCCA and TTR scenarios.

We realize OCCA and TTR scenarios in two different simulation environments, respectively, that have slightly different dynamics; we utilize IPG Automotive's CarMaker for OCCA, which is one of the most realistic simulation tools with sophisticated and accurate vehicle dynamics, and we utilize CARLA simulator (Dosovitskiy et al., 2017) for TTR, which is one of the most widely used simulation tools but does not have sophisticated and accurate vehicle dynamics.

Table 1: Success rate for 500 episodes of test runs for OCCA.

| METHOD | SUCCESS RATE | SUCCESS | FAIL |
|---|---|---|---|
| QC-SAC (Proposed) | 81.8% | 409 | 91 |
| BC-SAC (Lu et al., 2023) | 34.6% | 173 | 327 |
| SAC (Haarnoja et al., 2019) | 19.0% | 95 | 405 |
| BC | 0.0% | 2 | 498 |

Table 2: Average lap accomplishment rate and best lap time for 300 episodes of test runs for TTR.

| METHOD | AVERAGE LAP ACCOMPLISHMENT RATE | BEST LAP TIME |
|---|---|---|
| QC-SAC (Proposed) | 100.0% | 39.8s |
| BC-SAC (Lu et al., 2023) | 82.3% | 46.1s |
| SAC (Haarnoja et al., 2019) | 0.9% | - |
| BC | 5.5% | - |

## 4.1 OCCA IN CARMAKER

To develop a driving policy that can handle OCCA scenario, we first build a dataset from demonstrations of a human driver with a racing wheel input device. A total of 25,567 time-steps, equivalent to 21.3 minutes of driving data, are collected from 200 episodes. Using the collected data, we develop driving policies using four techniques: BC, SAC, BC-SAC, and QC-SAC. The resulting reward graphs per training episode are shown in Figure 4a. Additionally, we infer the driving policies and conduct 500 episodes of test runs to compare the control & avoidance success rates of each approach. During the test runs, the location of obstacles and the intensity and direction of the oversteer induced by the kick plate are set completely random. The results are shown in Table 1.

The reward graph and success rate of BC, which only learns from demonstration data without any interaction with the environment, testify the quality of the given demonstrations. As shown in Figure 4a, it is impossible to develop a good driving policy by solely using the given immature demonstration data. SAC, which maximizes rewards through the interaction with the environment, shows higher rewards than BC. However, as it is very difficult to experience successful episodes through

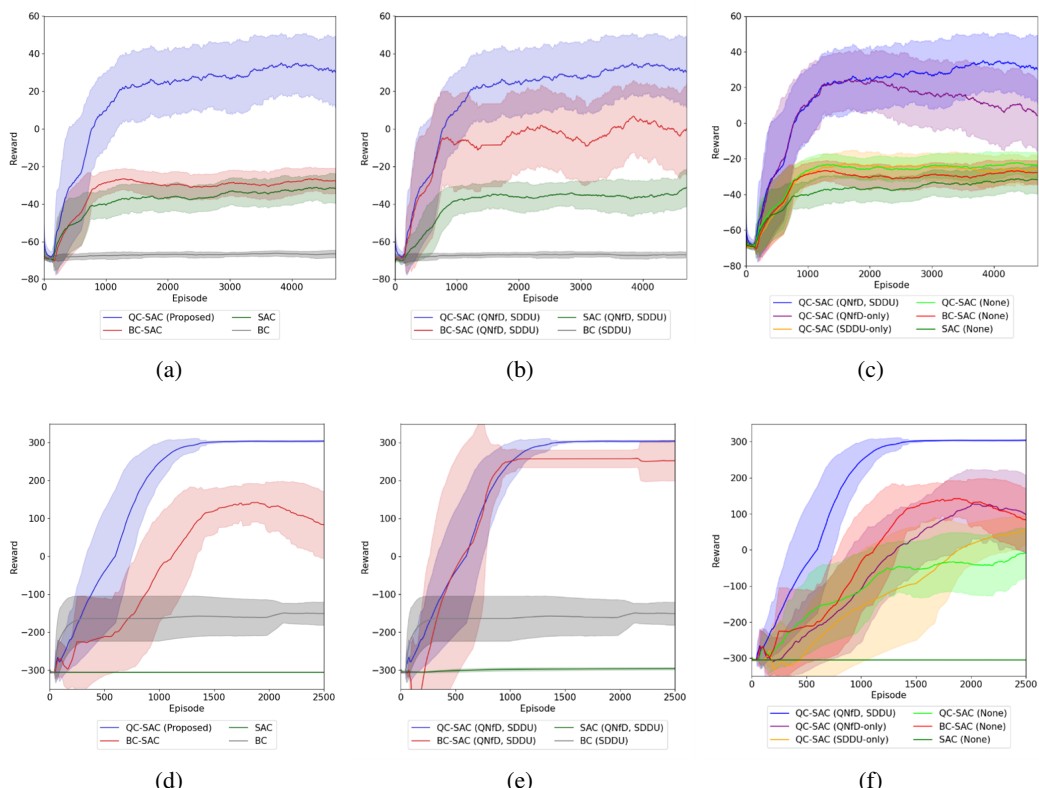

Figure 4: Training curve. The solid line represents the average reward of five instances of each training technique initialized with random seeds, while the shaded area indicates their standard deviation. The proposed QC-SAC achieves significantly higher rewards compared to other techniques in both scenarios. (a) Performance evaluation in OCCA, and (b,c) ablation study in OCCA. Because of the random obstacle placements, there are cases when the autonomous driving agent may not have any possible path to avoid the obstacles and the optimal policy inevitably experiences collisions. As a result, the standard deviation of the QC-SAC still remains high in the converged state. (d) Performance evaluation in TTR and (e,f) ablation study in TTR. The driving policy developed with QC-SAC perfectly converges with very low standard deviation. An optimal policy can have a standard deviation close to zero, since the TTR scenario has a fixed track environment without random obstacle placements. On the contrary, other training techniques fail to reach the optimum level, and they fail to complete the lap many times, resulting in high standard deviations.

random actions in the OCCA scenario, which has broad state and action spaces and high transition variance, SAC fails to develop the optimal driving policy. Note that selecting one wrong action in oversteer situation severely destabilizes the vehicle, making it impossible to complete the episode successfully. The probability that SAC consistently outputs appropriate actions at every time-step within an episode through random exploration is extremely low. BC-SAC, which considers both demonstration data and the interaction with the environment, performs slightly better than SAC but still cannot achieve high rewards as it continuously considers the immature demonstration data that disturbs the training.

In contrast, QC-SAC, which can effectively utilize immature demonstration data, records significantly higher rewards than other representative techniques. In the result of the test runs shown in Table 1, the proposed technique records a control & avoidance success rate of about 81.8%. While this success rate might seem insufficient in the autonomous driving field, where safety is critical, it represents a significant improvement given the 34.6% success rate of existing techniques and the challenges human drivers encounter in the same situation. Indeed, human drivers who collected the dataset record about 15% success rate when conducting 100 test runs, when the positions of the front

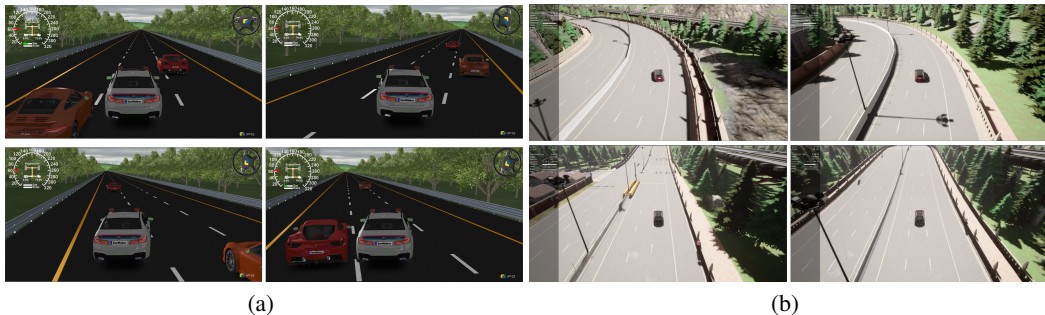

(a)             (b)

Figure 5: Captured scenes during the test runs. a, Videos of the test runs using QC-SAC in the OCCA scenario can be found at `https://youtu.be/j7Xr52TfEIQ`. b, Videos of the test runs using QC-SAC in the TTR scenario can be found at `https://youtu.be/rcEzif0PpQo`.

obstacles or the intensity and direction of the kick plate are unknown. Additionally, among the 91 episodes where QC-SAC fails to avoid collision, 95.6% (87 episodes) involve obstacles completely blocking the path in the direction of the vehicle's skid caused by the kick plate, making collision avoidance physically impossible within the given action space. Examples of these unavoidable collision failure cases can be seen in the video linked in Figure 5a. Therefore, excluding unavoidable collisions, the driving policy developed via QC-SAC achieves a 99.0% success rate, demonstrating an almost optimal (i.e., near-optimal) performance.

## 4.2 TTR IN CARLA

To develop a driving policy that can handle TTR scenario, we first build a dataset from demonstrations of a human driver by accumulating a total of 28,057 time-steps, which is equivalent to 23.4 minutes of driving data over 63 episodes. The demonstration data includes various immature driving actions such as collisions with barriers, driving at low speeds, skidding beyond the frictional limit, stopping during the drive, and unnecessary swerving. Both the reward graph shown in Figure 4d and the comparison of average lap accomplishment rates and best lap times shown in Table 2 demonstrate the superiority of QC-SAC over other representative techniques, i.e., BC-SAC, SAC, and BC. The average lap accomplishment rate refers to the mean percentage of track progress achieved, measured over 300 test runs. The reward graph of BC in Figure 4d testifies the quality of given immature demonstration data. Note that SAC performs worse than BC, failing to develop an effective action policy in all episodes. This is because TTR has a standing-start setup and continuous input on the throttle is required for a number of time steps to achieve a high speed. Therefore, as SAC attempts random actions for exploration (i.e., randomly alternates between throttle and brake), SAC fails to sufficiently accelerate the test vehicle or often stops the vehicle after a short distance. BC-SAC can start the vehicle appropriately, because it obtains a basic driving policy from the demonstration data and records higher performance compared to BC and SAC. However, it converges to a lower performance level than QC-SAC due to the immature demonstrations used for the basic driving policy. On the contrary, the proposed QC-SAC effectively utilizes immature demonstrations and successfully develops the optimal action policy with higher rewards and lower standard deviation than BC-SAC. It completes all 300 test runs without a single failure and reduces the lap time by over 13.6% compared to the driving policy developed via BC-SAC.

## 4.3 ABLATION STUDY

An ablation study is conducted on the 3 core elements of QC-SAC (i.e., QCO, QNfD, and SDDU) for both scenarios, OCCA and TTR. First, to evaluate the impact of the objective function, QNfD and SDDU are applied to BC-SAC, SAC, and BC, under all other conditions to be the same except for the objective function. Note that QNfD is not applicable to BC because it does not use Q values for training. The comparison results are shown in Figure 4b and 4e. Compared to Figure 4a and 4d, it can be seen that the reward of BC-SAC and SAC slightly improves due to the application of QNfD and SDDU, but still records significantly lower rewards compared to the proposed QC-

SAC. This confirms that QCO is essential for developing an optimal action policy when immature demonstrations are given.

For the evaluation of QNfD and SDDU, we remove each function from the QC-SAC and observe the performance degradation. The results with SDDU and QNfD removed from the QC-SAC are labeled as QNfD-only and SDDU-only, respectively, while the results with both methods removed are labeled as None. As shown in Figure 4c and 4f, performance declines when SDDU and QNfD are not used, especially when QNfD is removed, there is a significant performance degradation. Because QC-SAC evaluates demonstration data based on Q values, when the performance of the Q-network decreases as the demonstration data is not used in Q-network training, the overall QC-SAC shows a substantial performance drop. This proves that QNfD is essential for QCO. On the other hand, when SDDU is discarded from QC-SAC, the reward graph starts to decline after around 2,000 episodes of training. This can be expected due to the overfitting from continuous behavior cloning on a fixed small set of demonstration data. Among the collected 25,567 and 28,057 time-steps of data for each scenario, excluding those data with low Q values that are not considered by the QCO, a small amount of data could have been used for training, which causes an overfitting. In contrast, QC-SAC, which continuously updates and improves the demonstration data with successful episodes, does not encounter overfitting problem. Lastly, when both QNfD and SDDU are not implemented into QC-SAC, the performance of QC-SAC degrades significantly. However, due to the significant degradation by the removal of QNfD, the difference between None and QNfD-only is not substantial.

The results of the ablation study prove that all of the three core elements of QC-SAC, QCO, QNfD, and SDDU, are essential and critical. Therefore, the best performance is achieved when all of these three elements are employed.

## 5 CONCLUSION

In this study, we have introduced a safety-advanced autonomous driving technology using Q-Compared Soft Actor-Critic (QC-SAC), for urgent hazardous situations (UHS) that are difficult to cope with even for expert human drivers. In such situations, developing a driving policy using existing IL, RL, or HL techniques is challenging because RL relies on random exploration to experience and collect data from successful episodes, which is unlikely in UHS, and IL and HL require a variety of optimal expert demonstrations that are nearly impossible to obtain. The proposed QC-SAC is an innovative HL technique that can effectively utilize immature demonstration data and adapt quickly to novel situations to develop an optimal driving policy. We have demonstrated the superior performance of the QC-SAC for two extreme UHS scenarios, oversteer control with collision avoidance (OCCA) and time-trial race (TTR), to the representative and conventional techniques, such as BC, SAC, and BC-SAC. It has been found that QC-SAC is the world-first safety-advanced driving policy that can avoid obstacles ahead while controlling the oversteer, demonstrating that QC-SAC is the first solution to prevent unexpected fatal accidents due to the UHS.

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

# A  APPENDIX

## A.1  PRIOR WORKS: IL, RL, AND HL

IL is a form of supervised learning that aims to imitate the actions of an expert. The objective of IL is generally expressed as follows:

$$\arg \min_{\phi} \mathbb{E}_{(s_E, a_E) \sim \mathcal{D}}[\mathcal{L}(\pi_\phi(s_E), a_E)], \tag{9}$$

where $\mathcal{D}$ is the dataset collected from the expert demonstration, $s_E$ is the state experienced in $\mathcal{D}$, $a_E$ is the expert's action in $s_E$, $\pi_\phi$ is the IL action policy parametrized by $\phi$ that we are training, and $\mathcal{L}$ is the loss function. The action policy $\pi_\phi$ is trained to output actions similar to $a_E$ for all $s_E$. However, due to the nature of supervised learning, $\pi_\phi$ can output actions similar to the expert actions only for the states or the similar states to those states included in the dataset. Therefore, when there is a shortage of expert data, or the data is immature, significant performance degradation is inevitable.

In contrast, RL is a technique where the agent interacts with the environment to develop an action policy that can maximize the cumulative rewards. It figures out appropriate actions to maximize the cumulative rewards on each state, based on its own experiences evaluated with the predefined reward function. The objective of RL is generally formulated as follows:

$$\max_{\phi} \mathbb{E}_{(s_t, a_t) \sim \rho_{\pi_\phi}} \left[ \sum_{t=0}^{\infty} \gamma^t r(s_t, a_t) \right], \tag{10}$$

where $\pi_\phi$ is the RL action policy to be trained parametrized by $\phi$, $\rho_{\pi_\phi}$ is the distribution of trajectory experienced by the agent using $\pi_\phi$, $r(s_t, a_t)$ is the reward at time-step $t$, and $\gamma$ is the discount factor.

Consequently, RL is trained to maximize the expectation of cumulative rewards obtained along an episode. Because RL experiences various episodes through extensive trial and error, it has the advantage of being robust in various situations compared to IL. However, as it needs to be trained through numerous interactions with random actions, it has low sample efficiency and slow training speed. Especially, the more complex the task, the broader the state space and action space, and the higher the transition variance, the more these problems are exacerbated, making it difficult to develop the optimal action policy or even initiate the training.

To complement the weaknesses of both IL and RL mentioned above, recent research on HL has focused on combining the two techniques. There are three main approaches to this fusion. The most representative approach is using pre-training and fine-tuning (Hester et al., 2018; Rajeswaran et al., 2017; Tian et al., 2021; Cai et al., 2021; Wang & Chang, 2019), where we initialize the weights of the policy network using IL (i.e., pre-training) and then fine-tunes the policy network using RL. Although this is the simplest approach, when the demonstration data for IL is immature, the action policy developed through IL can diverge significantly from the optimal policy pursued by RL. Consequently, during the fine-tuning process, the model weights may change drastically, potentially causing training instability, and making the pre-trained IL model ineffective.

The second approach is the residual RL, which first train an IL-based action policy network over a demonstration data and then train a residual RL-based action policy network to find improvements in the IL-based action policy (Alakuijala et al., 2021). The agent executes the numerical sum of the actions from the IL-based and residual RL-based policies. This approach can alleviate the training instability problem that arises in the pre-training and fine-tuning approach, by utilizing separate

networks for IL and RL. However, since it simply uses the added actions from IL and RL, it cannot complement the shortcomings of IL in cases of immature demonstrations or data scarcity.

The final approach combines the objectives of IL and RL so that demonstration data is considered together when training RL. A representative example is Behavior Cloned Soft Actor-Critic (BC-SAC) (Lu et al., 2023), where the objective function of Behavior Cloning (BC) is added to the objective function of Soft Actor-Critic (SAC) (Haarnoja et al., 2018) as:

$$\max_{\phi} \mathbb{E}_{(s,a)\sim\rho_{\pi_\phi}}[Q(s,a) + \alpha\mathcal{H}(\pi_\phi(\cdot|s))] + \lambda \cdot \mathbb{E}_{(s_E,a_E)\sim\mathcal{D}}[\log\pi_\phi(a_E|s_E)], \quad (11)$$

where $Q(s,a)$ is the action value function, $\mathcal{H}$ is the entropy, $\alpha$ is the temperature, which is the weight of the entropy term, $\lambda$ is the weight of the BC objective, and the first expectation term is the objective function of SAC. As described earlier, BC-SAC uses the demonstration data for RL rather than using a separate IL network trained with the demonstration data. Because of this, BC-SAC is robust even when the demonstration data is scarce, but the limitation is that the given demonstration must contain optimal data. Notice that the BC term, i.e., the second term in (11), tries to train the policy network to produce similar actions to the given demonstration, even if the demonstration is immature, while the SAC objective function directs the policy network to generate actions to maximize the cumulative reward. Therefore, when the given demonstration is noisy and does not contain optimal demonstration, these two objectives interfere each other and deteriorate the training process, as represented in Figure 6.

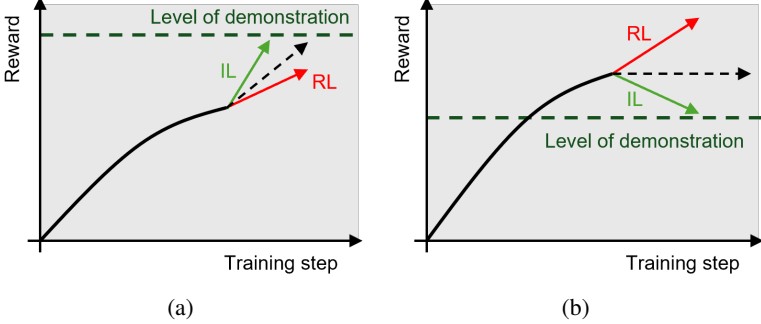

(a)   (b)

Figure 6: Concept diagram. Impact of the quality of demonstration data on the training of action policy using existing HL techniques. (a) IL helps the training, when optimal demonstration data is given. (b) IL deteriorates the training, when immature demonstration data is given.

## A.2   EXPERIMENTAL SETUP - OCCA

### A.2.1   SCENARIO SETUP

To compose the OCCA scenario, a kick plate and obstacles are configured in the simulator. First, to deliberately induce oversteer, road friction is reduced by 50%, and a virtual kick plate is configured to apply lateral force to the rear tires as the vehicle passes a certain point. The lateral force applied to the rear tires is randomly set to vary the intensity and direction of the oversteer. Additionally, as shown in Figure 3c, obstacles are randomly placed within 30 to 70 m distance ahead of the kick plate. The distance between the kick plate and the obstacles is set closer than the legally recommended safe distance[1], rendering simple braking insufficient for collision avoidance and necessitating the formulation of precise avoidance trajectories. Furthermore, to prevent scenarios where the road is entirely obstructed and driving becomes unfeasible, the number of obstacles placed is one fewer than the number of lanes. In the experiments, up to two obstacles are randomly positioned on a three-lane road.

---

[1]Converting the distance between the kick plate and the obstacles to Time to Collision (TTC) at an entry speed of 70 km/h, 30 m corresponds to 1.54 s, and 70 m corresponds to 3.60 s, which are within the legally defined safe distances. Generally, the safety distance required is 2 s in the US and Europe and 3.6 s in South Korea.

A.2.2 STATE SPACE

The state space serving as input for RL and IL models is divided into two categories: vehicle state and surrounding state. The vehicle state includes eight values: side slip angle ($\beta$), longitudinal velocity ($v_{long}$), longitudinal acceleration ($a_{long}$), lateral acceleration ($a_{lat}$), cross-track error from original path or lane ($d$), orientation error with respect to original path ($\psi$), yaw rate ($\dot{\psi}$), and steering angle ($\delta$). Side slip angle is calculated from $v_{long}$ and lateral velocity ($v_{lat}$) ($\beta = \arctan\left(\frac{v_{lat}}{v_{long}}\right)$), while $d$ and $\psi$ are calculated based on the center line of the lane in which the vehicle was traveling before oversteering and the vehicle's center of gravity, respectively (see Figure 7a). The reason for considering the steering angle in the vehicle state will be discussed in the next section, with the action space setup. The surrounding state is composed of a 90-dimensional vector. This is a vectorized drivable area within a 90° azimuth range centered to front direction of the vehicle with 1° resolution, representing the drivable distance for each azimuth angle in polar coordinates. For each azimuth angle, the shortest distance to an obstacle or the road boundary is recorded. The visualization of the surrounding state vector can be seen in Figure 7b.

A.2.3 ACTION SPACE AND ACTION CONSTRAINT

The action space is represented with a two-dimensional vector of pedal value and steering angular velocity. Generally, in end-to-end autonomous driving systems, the output action consists of longitudinal and lateral control values. For longitudinal control, it is typically either the target speed (Tian et al., 2021) or the pedal value (Wurman et al., 2022; Cai et al., 2020; Cutler & How, 2016; Wang & Chang, 2019), while the lateral control value is usually the steering angle (Wurman et al., 2022; Tian et al., 2021; Cai et al., 2020; Cutler & How, 2016; Wang & Chang, 2019). In the oversteer situation considered in this study, changes in weight transfer or slip ratio caused by pedal manipulation are more critical than the vehicle's speed. Therefore, the pedal value is selected as the longitudinal control value instead of the target speed. For lateral control, steering angular velocity is used instead of the steering angle to constrain the steering angular velocity. Note that when the steering angle is chosen as an action output, we cannot constrain the steering angular velocity, as the policy network can output very different steering angles at two consecutive time-steps. Indeed, when we develop a driving policy to produce the steering angle as an action, the maximum steering angular velocity records 18,000°/s. That means when the simulation runs at 20 Hz and the test vehicle has a steering range from -450° to 450°, a driving policy that steers with the maximum angular velocity is trained. An experiment video can be found at `https://youtu.be/gKSK9DkzRCY`. Since it is impractical to control the steering wheel in such a high angular velocity, we propose a method to constrain the vehicle's steering angular velocity. By specifying the steering angular velocity ($\dot{\delta}$) as an action output and considering the steering angle ($\delta$) as part of the state, the steering angular velocity can be constrained without violating the Markov property. Considering the Markov property that the current state $s_t$ is influenced only by the previous state $s_{t-1}$ and the previous action $a_{t-1}$, and the definition that $\delta_{t-1} \in s_{t-1}$ and $\dot{\delta}_{t-1} \in a_{t-1}$, $\delta_t \in s_t$ can be expressed as $\delta_t = \delta_{t-1} + \dot{\delta}_{t-1} \cdot \Delta t$. Since time step $\Delta t$ is fixed at 0.05 s in our 20 Hz simulation setup, our method does not violate the Markov property. In the simulation, both outputs, pedal value and steering angular velocity, are normalized between -1 and 1. Negative and positive pedal values indicate the brake and throttle pedal manipulations, respectively. The steering angular velocity is set by multiplying the output value by 700°/s, ensuring that the maximum steering angular velocity is 700°/s. This value is based on the maximum steering angular velocity of the steer-by-wire system (Yih & Gerdes, 2005).

A.2.4 REWARD FUNCTION SHAPING

The reward function consists of four components: safe distance reward ($R_{safe}$), progress reward ($R_{prog}$), auxiliary reward ($R_{aux}$), and terminal reward ($R_{term}$). Except for $R_{prog}$ and $R_{term}$, all reward functions adhere to the form of

$$r_t(x) = 0.5^{|x|/\bar{x}} \times 2 - 1, \tag{12}$$

where $x$ is the value of interest, and $\bar{x}$ is the predefined requirement value for $x$. This form of reward function is suitable for situations where a smaller $x$ value is desired. It limits the range of the reward value and considers the requirement value. Starting from an exponential function $r_t(x) = e^{-k|x|}$, it

is modified to $r_t(x) = e^{-k|x|} \times 2 - 1$ to limit the reward value between -1 and 1. The x-intercept (at $x = -\frac{\ln(2)}{k}$) is set as the requirement value we aim to achieve at least. Aligning the x-intercept with the requirement value yields (12). The visualization of the reward function form is shown in Figure 7c.

Using the above function form in (12), each reward component considers the following variables as input $x$. For $R_{safe}$, the minimum value from the surrounding state information (drivable distance expressed in polar coordinates) is considered as the input. To maximize the safety distance from obstacles, -1 is multiplied when summing the total reward. For $R_{aux}$, the cross-track error, side slip angle, and steer rate are considered. We aim to minimize these values, in order to enhance passenger comfort, and improve control stability. Also, we strive to find an avoidance path that does not deviate significantly from the original path (i.e., the path planned before the overstreer) whenever possible.

Moreover, $R_{prog}$ considers the distance traveled in the tangent direction within the Frenet-Serret frame, with respect to the path planned before the oversteer. The $R_{prog}$ at time-step $t$ is calculated as the difference in the distance traveled between time-steps $t$ and $t-1$. Finally, $R_{term}$, added at the end of each episode, evaluates whether the episode was successful. When the side slip angle remains stable less than 1° for 100 time-steps, we consider that the vehicle is in the grip state, and the episode ends, granting a reward of +50. When the vehicle goes off-road, collides with an obstacle, or the side slip angle exceeds 37° (the vehicle's maximum wheel steer angle) and spins, a penalty of -50 is given.

The total reward ($R_{tot}$) that combines all four reward components is defined as follows:

$$R_{tot} = -\lambda_1 R_{safe} + \lambda_2 R_{prog} + \lambda_3 R_{aux} + R_{term}, \tag{13}$$

where $\lambda_i$ for $i \in \{1, 2, 3\}$ is the weighting factor for each reward term.

In this study, we use weight values of $\lambda_1 = 0.8$, $\lambda_2 = 0.2$, $\lambda_3 = 0.2$. Additionally, the requirement values are set as follows: cross-track error to 3.5 m (the width of one lane), acceleration to 2.943 m/s$^2$, side slip angle to 20°, and steer rate to 3000°/s.

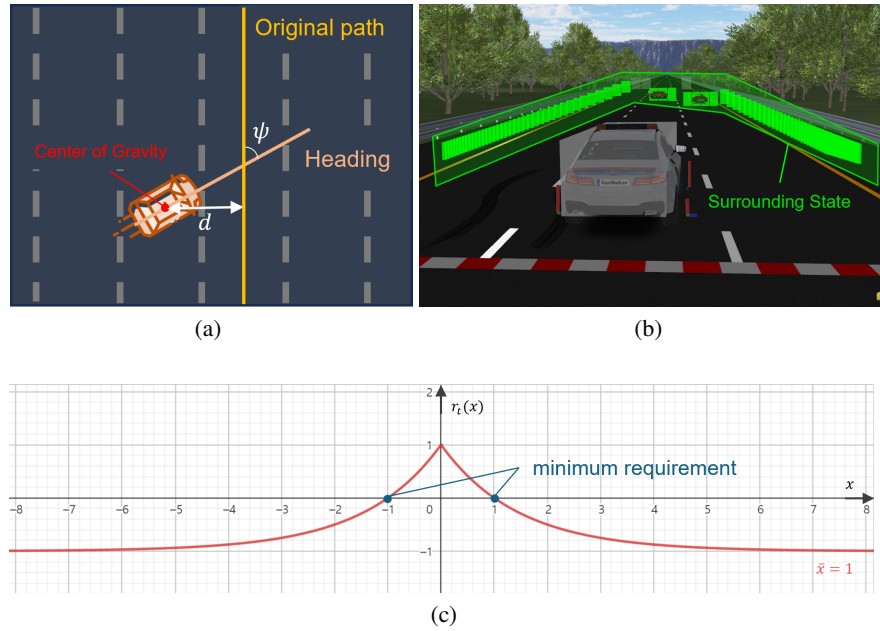

(a)    (b)

(c)

Figure 7: Experimental setup. (a) Definition of $d$ and $\psi$ in vehicle state. (b) Representation of surrounding state. (c) An example of the reward function. ($\bar{x} = 1$).

### A.3 EXPERIMENTAL SETUP - TTR

As aforementioned, the goal of the TTR is to complete driving along a predetermined track in the earliest time possible. In this study, we set up a scenario with the objective of fast driving on the outer highway of Town05 in the CARLA simulator. The outer highway of Town05 includes four 90° turns and significant elevation changes that can cause the vehicle to lose grip at high speeds, making it a challenging environment. The state space and action space are the same to the OCCA scenario. The reward function is also the same to that introduced in the previous section, with an additional reward $R_{speed}$ aimed at pursuing high speeds, more suitable for the race task. The reward $R_{speed}$ follows the form of (12), with a requirement value of 20 m/s. All other requirement values remain the same to those introduced earlier. Additionally, the terminal condition is changed from maintaining a low side slip angle to completing the track. The final reward function is as follows:

$$R_{tot} = -\lambda_1 R_{speed} - \lambda_2 R_{safe} + \lambda_3 R_{prog} + \lambda_4 R_{aux} + R_{term}, \tag{14}$$

where $\lambda_i$ for $i \in \{1, 2, 3, 4\}$ is the weighting factor for each reward terms. The weights for each reward term are set as $\lambda_1 = 2.0$, $\lambda_2 = 0.8$, $\lambda_3 = 0.2$, and $\lambda_4 = 0.2$.

### A.4 QC-SAC ALGORITHM PSEUDOCODE

---
**Algorithm 1** Q-value Compared Soft Actor-Critic (QC-SAC)

---
**Require:** demonstration dataset $\mathcal{D}$, average episode reward of demonstration $\bar{r}_{epi}$, discount factor $\gamma$, target update rate $\tau$, target entropy $\bar{\mathcal{H}}$, and learning rates $\lambda_Q, \lambda_\pi, \lambda_\alpha$
1: Initialize policy network $\pi_\phi$ and Q networks $Q_{\theta_1}$, $Q_{\theta_2}$ with random weights
2: Initialize target Q-networks $Q_{\bar{\theta}_1}^-$, $Q_{\bar{\theta}_2}^-$ with weights $\bar{\theta}_1 \leftarrow \theta_1, \bar{\theta}_2 \leftarrow \theta_2$
3: Initialize entropy coefficient $\alpha$
4: Initialize replay buffers $\mathcal{D}_{RL} \leftarrow \emptyset, \mathcal{D}_{epi} \leftarrow \emptyset$
5: Initialize episode reward $r_{epi} \leftarrow 0$
6: **for** each iteration **do**
7:     **for** each environment step **do**
8:         $a \sim \pi_\phi(a|s)$
9:         $s' \sim p(s'|s, a)$
10:         $\mathcal{D}_{RL} \leftarrow \mathcal{D}_{RL} \cup \{(s, a, r, s', d)\}$
11:         $\mathcal{D}_{epi} \leftarrow \mathcal{D}_{epi} \cup \{(s, a, r, s', d)\}$
12:         $r_{epi} \leftarrow r_{epi} + r$
13:         **if** $d = 1$ **then**
14:             **if** $r_{epi} > \bar{r}_{epi}$ **then**
15:                 $\mathcal{D} \leftarrow \mathcal{D} \cup \mathcal{D}_{epi}$
16:                 $\bar{r}_{epi} \leftarrow (\bar{r}_{epi} \cdot (|\mathcal{D}| - 1) + r_{epi}) / |\mathcal{D}|$
17:             **end if**
18:             $r_{epi} \leftarrow 0$
19:             $D_{epi} \leftarrow \emptyset$
20:         **end if**
21:     **end for**
22:     **for** each gradient step **do**
23:         Sample $\mathcal{B}_{RL} = \{(s, a, r, s', d)\}$ from $\mathcal{D}_{RL}$
24:         Sample $\mathcal{B}_{BC} = \{(s_d, a_d, r_d, s'_d, d_d)\}$ from $\mathcal{D}$
25:         Combined batch $\mathcal{B} = \mathcal{B}_{RL} \cup \mathcal{B}_{BC}$
26:         $\theta_i \leftarrow \theta_i - \lambda_Q \nabla_{\theta_i} J_Q(\theta_i)$ for $i \in \{1, 2\}$ using $\mathcal{B}$
27:         $\phi \leftarrow \phi - \lambda_\pi \nabla_\phi J_\pi(\phi)$ using $\mathcal{B}_{RL}, \mathcal{B}_{BC}$
28:         $\alpha \leftarrow \alpha - \lambda \nabla_\alpha J(\alpha, \bar{\mathcal{H}})$ using $\mathcal{B}_{RL}$
29:         $\bar{\theta}_i \leftarrow \tau \theta_i + (1 - \tau)\bar{\theta}_i$ for $i \in \{1, 2\}$ using $\mathcal{B}$
30:     **end for**
31: **end for**

---

