# OpenReview forum: "Safety-Advanced Autonomous Driving for Urgent Hazardous Situations using Q-Compared Soft Actor-Critic"
_ICLR.cc/2025/Conference — ICLR 2025 Conference Withdrawn Submission_

### Official Review · Reviewer_qYs1 · 2024-10-31

**Soundness:** 3
**Presentation:** 2
**Contribution:** 3
**Rating:** 3
**Confidence:** 4

**Summary:**

**Overview:** This paper proposes a method for training a self-driving policy to overcome urgent hazardous situations (UHS). Though human demonstration is used, unlike directly applying BC+SAC authors propose a new algorithm, enabling the vehicle to recover from oversteering while avoiding obstacles. The method is also used to train driving policy in CARLA. In both experiments, the proposed method outperforms SAC/SAC+BC/BC baselines by a large margin, demonstrating the effectiveness of the method.

**Method:** A human-demonstrated dataset is first collected in the target scenario, and then SAC agents are trained in the same scenario with demonstration data. In this paper, an oversteering scenario is created in the simulator and serves as the training environment.

**Strengths:**

1. Training policies to find the optimal solutions in systems with complicated dynamics is difficult. This paper proposes a simple method, allowing the agent to do efficient exploration with some human demonstrations.
2. The method shows superior performance in two experimental settings, compared to other baselines.

**Weaknesses:**

The main weakness of this paper is that the generalizability of the method remains unknown. Incorporating a human-demonstrated dataset into RL training is very common with many existing solutions, so I believe the proposed method should be tested in other settings besides autonomous driving i.e. manipulation task, and compared with other methods besides naive SAC+BC. As the method doesn't require a task-specific algorithm/reward design, I believe it has such potential.

As a learning-conference, ICLR values the generalizability of the method. Thus, I would expect it to work well in many standard benchmarks first (Though the RL benchmark is not as standard as those used by CV people) and the oversteering setting only serves as an interesting case study.  In addition, the paper writing should be adjusted to cover a wide scope not limited to USH for car driving. If this method remains a solution for a certain kind of problem (vehicle oversteering/understeering), ICRA/ITSC would be more suitable for this manuscript.

**Questions:**

N/A

---

### Official Review · Reviewer_onJa · 2024-11-02

**Soundness:** 2
**Presentation:** 2
**Contribution:** 1
**Rating:** 3
**Confidence:** 4

**Summary:**

In this paper, the authors propose a learning method that mixes RL and IL to handle urgent hazardous situations in autonomous driving. The method proposes to mix the learning target of IL and RL and introduce several learning tricks to improve the performance of the learned policy. The policy is changed to optimize the SAC target plus the weighted L1 norm BC target. The authors also propose QNfD and SDDU to help learn the accurate Q-value during the learning. The proposed method is evaluated in two urgent hazardous situations: OCCA and TTR in simulation and compared with baselines including BC, SAC, and BC-SAC.

**Strengths:**

The paper is easy to read and follow. The paper has clear plots that show the result.

**Weaknesses:**

The first concern is whether the proposed method has been fairly evaluated with the baselines. The title and main introduction suggest the focus is solving the urgent hazardous situations in autonomous driving. With that as the focus, only 3 general learning-based methods are compared as baselines, which may not stand as a fair comparison. The experiments should provide a comparison between SOTA methods for this domain rather than those basic methods. For example, for the second evaluated task: the time-trial race, there are lots of different methods in the literature that can solve the problem well, for example [1-3]. Those methods also proposed their own "learning tricks" to solve the problem. It is essential to provide a comparison to show why the proposed add-ons to the SAC are better than those in the literature.

The second concern is about the limitation of evaluated scenes. The current experiments are limited in cases. For the OCCA, there could be different cases in which oversteering may happen. The proposed method only trained in one scene with some randomization on obstacles. There is no evaluation of different scenes. For example, now the scene is on a straight road, what will happen if it is on a curve or with a different width or more obstacles? Do I need to train the policy over all possible scenes? Will it be possible to encode that map info and build a general policy that works on novel scenes? As a method for solving OCCA, it is essential to answer those questions. Otherwise, it would be more like a benchmark task for evaluating learning methods rather than evaluating for solving the task.

With these two mentioned concerns, it is worth noting that as a paper for handling urgent hazardous situations in autonomous driving, the paper lacks sufficient problem formulation and system modeling. Section 2 is more like a general introduction rather than a mathematical problem formulation and modeling. The main method in the paper is also not a domain-specific method. Therefore, the whole structure of the paper is more like proposing a generic learning method for solving hard tasks rather than focusing on solving the urgent hazardous situations in autonomous driving. If the author presents the work as a generic learning method for solving hard tasks, the current baseline result with one or two more SOTA IL+RL methods [4-7] may be enough in some sense, and the current scenes in OCCA and TTA are also enough to show the ability of the proposed method in the driving domain. However, if that is the focus, the experiments should include different applications from different domains, and the title and whole introduction should be changed, which I don't think is a feasible track for the current submissions. But the current way of presenting is definitely not a good fit.

The third concern is about theoretical soundness. The proposed method uses eq. (5) and eq. (6) to learn the critic. However, these equations are from SAC and are only correct if the policy is the target policy of SAC which is the max-entropy policy. Since the learning target of the current policy has been changed to a mix of the SAC target with a BC target, the math has changed. Thus, eq. (5) and eq. (6) are not the Q function for the current optimization target. It is OK to have a biased learning target. But it is necessary to discuss the reason why we would like to do that, what is the benefit, and what is correct math. Meanwhile, the BC target needs Q to obtain the weight. It is clear that at the beginning the Q net is not accurate and thus the weight would also be wrong. It is also important to discuss if this will cause the training unstable and why. However, the paper misses those discussions about math and proof.

The last concern is about the novelty. The proposed method is pretty similar to the BC-SAC method. The method in this paper just changes the learning target of the added BC target and adds some learning tricks (QNfD, SDDU). The current results also show that the performance of BC-SAC and QC-SAC are pretty similar. They both perform similarly well with QNfD and SDDU on and similarly bad with QNfD and SDDU off. These do show the contribution of the proposed QNfD and SDDU. But I feel the main methods are still the same (BC-SAC also has the theoretical soundness concern as I mentioned above).

With all these concerns, I think the author probably needs to rethink their way of presenting the method and probably go for another opportunity to publish.

Here is the summarized list of the missed content:

1. The experiment lacks enough baseline methods especially those SOTA methods in the focused task, for example, SOTA in the autonomous racing domain.
2. The experiment lacks enough scenarios, including different OCCA scenes and discussion about the performance in novel scenes and different racing tracks for the TTA tasks, which makes the evaluation more like a benchmark task for evaluating general RL IL learning methods rather than truly evaluating if the focused task is solved.
3. The experiment lacks enough detail about parameters for the training and demonstration dataset, e.g. for OCCA, are they all successful, or do they only contain the oversteer part? It is hard to say whether the poor result of the baseline methods is due to parameter tuning or the method itself, especially given sufficient papers in the literature showing SAC does solve the TTR task.
4. The method lacks theoretical soundness. As mentioned in the above, the equation seems to be wrong or biased in theory. The weight also depends on Q net which is definitely biased at the beginning. However, there is no discussion or proof about those details.
5. The method lacks enough novelty. Compared with BC-SAC, the method in this paper just changes the learning target of the added BC target and adds some learning tricks (QNfD, SDDU).

[1] Wurman, Peter R., et al. "Outracing champion Gran Turismo drivers with deep reinforcement learning." *Nature* 602.7896 (2022): 223-228.

[2] Vasco, Miguel, et al. "A Super-human Vision-based Reinforcement Learning Agent for Autonomous Racing in Gran Turismo." REINFORCEMENT LEARNING CONFERENCE (RLC) (2024).

[3] Hao, Ce, et al. "Skill-Critic: Refining Learned Skills for Hierarchical Reinforcement Learning." *IEEE Robotics and Automation Letters* (2024).

[4] Hester, Todd, et al. "Deep q-learning from demonstrations." *Proceedings of the AAAI conference on artificial intelligence*. Vol. 32. No. 1. 2018.

[5] Nair, Ashvin, et al. "Awac: Accelerating online reinforcement learning with offline datasets." arXiv preprint arXiv:2006.09359 (2020).

[6] Xue, Zhenghai, et al. "Guarded policy optimization with imperfect online demonstrations." *arXiv preprint arXiv:2303.01728* (2023).

[7] Li, Chenran, et al. "Residual q-learning: Offline and online policy customization without value." Advances in Neural Information Processing Systems 36 (2023): 61857-61869.

**Questions:**

My questions and suggestions would be the same as the described weakness.

1. Adding more baselines. It is essential to provide a comparison to show why the proposed add-ons to the SAC are better than those SOTAs in the literature.
2. Adding more scenarios to show the proposed method does solve the real QCCA problem rather than showing a demo in one example scene.
3. Adding more details about the training parameters, demonstration collection, and demonstration distribution.
4. Adding discussions and necessary proof regarding the method. Are eq.(5,6) consistent with the defined optimization eq.(1)? If they are consistent in theory, please provide the proof. If not, please provide the reason why we would like to do that and provide the details about what the current critic really learned. Meanwhile, as discussed in the summary, the weight C looks clearly biased at the beginning stage. Please provide the details about why it doesn't matter.

---

### Official Review · Reviewer_Ycpd · 2024-11-02

**Soundness:** 3
**Presentation:** 2
**Contribution:** 2
**Rating:** 5
**Confidence:** 4

**Summary:**

This paper introduces Q-Compared Soft Actor-Critic (QC-SAC), a novel hybrid learning technique for autonomous driving in urgent hazardous situations (UHS). QC-SAC addresses the limitations of existing imitation learning, reinforcement learning, and hybrid learning methods by effectively utilizing immature demonstration data to develop optimal driving policies and adapting quickly to new situations. The paper demonstrates QC-SAC's superior performance in two extreme UHS scenarios: oversteer control with collision avoidance (OCCA) and time-trial race (TTR). In OCCA, QC-SAC achieves a significantly higher success rate compared to existing techniques, and in TTR, it reduces lap time by over 13.6% without failures. The paper's contributions include proposing an innovative learning approach that enhances autonomous driving safety in challenging scenarios, providing the first technology capable of safely controlling vehicle oversteer and avoiding obstacles.

**Strengths:**

This paper introduces a novel hybrid learning technique, Q-Compared Soft Actor-Critic (QC-SAC), which addresses the challenge of uncertainty in autonomous driving, particularly in urgent hazardous situations. It stands out for its originality in combining information-theoretic approaches with imitation and reinforcement learning to effectively utilize immature demonstration data and adapt to new scenarios. The quality of the research is evident in the rigorous experimental design and the comprehensive analysis provided, demonstrating QC-SAC's superior performance over existing methods in both oversteer control with collision avoidance and time-trial race scenarios. The clarity of the paper is commendable, with a well-structured presentation of the methodology, experiments, and results, making complex concepts accessible. The significance of the work lies in its potential to enhance the safety and reliability of autonomous vehicles in high-risk situations, which is a critical advancement for the field.

**Weaknesses:**

While the paper presents a promising approach with QC-SAC for handling uncertainty in autonomous driving, a potential weakness lies in the generalizability of the findings. The experiments, although thorough within the two specified scenarios, may not fully capture the breadth of real-world conditions. To strengthen the paper, future work could expand the evaluation to include a more diverse set of hazardous situations and different environmental conditions, ensuring that the robustness of QC-SAC is not only limited to the tested scenarios but also applicable to a wider range of unpredictable driving contexts. Additionally, incorporating a comparison with state-of-the-art methods on standardized benchmarks would provide a clearer picture of how QC-SAC performs relative to existing solutions in the field.

**Questions:**

How does QC-SAC handle highly variable or noisy demonstration data? Can the authors provide insights into the algorithm's sensitivity to the quality of initial demonstration data?
The paper focuses on two specific UHS scenarios. How well does QC-SAC generalize to other types of hazardous situations not covered in the study?
How does QC-SAC balance exploration and exploitation, especially in scenarios with high state and action spaces?
Are there any recent state-of-the-art methods that were not included in the comparison? If so, how might QC-SAC compare to these methods?
For the 18.2% failure rate in OCCA, could the authors provide a deeper analysis of the failure cases? Are there common factors or specific conditions that lead to these failures?

---

### Official Review · Reviewer_RzqZ · 2024-11-04

**Soundness:** 2
**Presentation:** 2
**Contribution:** 2
**Rating:** 3
**Confidence:** 3

**Summary:**

The paper provides a new RL method QC-SAC for urgent hazardous situations (UHS), such as skidding or tire grip loss. This method utilizes immature demonstration data by evaluating the quality of this data based on action value Q, allowing the model to prioritize useful information while discarding less beneficial data. QC-SAC aims to enhance the performance of the Q-network and facilitate quick adaptation to new situations by integrating successful experiences from ongoing interactions.

**Strengths:**

S1. Introduces a new kind of problem to prevent UHS like oversteer and understeer in vehicles.

S2. Shows high success rate in the provided data set and the videos provided justify the results qualitatively.

**Weaknesses:**

W1. A clear methodology figure is missing which makes the paper difficult to follow.

W2. An explanation of the dataset used with the distribution of data, and the kinds of trajectories the dataset has is not provided. The testing dataset is also not explained.

W3. Intuition for using QC-SAC and its development ideology could be explained more clearly to help the readers understand the algorithm.

**Questions:**

Same as Weaknesses. Additional questions:

Q1. Could you please elaborate on W2 and also provide the distribution of UHS cases in the formed dataset.

Q2. It’s surprising to see that Figure 4(d) BC and Figure 4(e) BC (SDDU) have reward curves that look identical. Could the authors elaborate on that?

---

### Official Review · Reviewer_mGWj · 2024-11-05

**Soundness:** 2
**Presentation:** 2
**Contribution:** 2
**Rating:** 3
**Confidence:** 4

**Summary:**

This paper introduces Q-Compared Soft Actor-Critic (QC-SAC), a hybrid learning method designed for autonomous driving in urgent hazardous situations (UHS) such as skidding on slippery roads or maintaining control at high speeds, where collecting optimal demonstration data or using RL alone could be difficult. QC-SAC builds upon imitation learning (IL) and reinforcement learning (RL) but addresses their limitations in high-stakes, complex UHS by prioritizing high-quality, immature demonstration data through a Q-value comparison mechanism. The approach integrates demonstration data effectively with RL updates to learn policies, demonstrating robust performance in oversteer control with collision avoidance and time-trial race scenarios.

**Strengths:**

* The proposed method merges imitation and reinforcement learning, combining both into a single objective function without requiring a separate pre-training stage.
* It selectively prioritizes high-quality policy rollout data to complement the original demonstrations, thereby providing more diverse data for behavior cloning.

**Weaknesses:**

* The proposed method is only evaluted in limited scenarios for urgent hazardous situations of autonoumous driving.
* The novelty of the paper is unclear. The main contribution appears to be the integration of selected demonstration data into SAC’s Q-function updates and the use of a weighted behavior cloning loss to supplement the SAC policy update. Could the authors further clarify these contributions?
* Some statements regarding SAC seem too strong or not accurate, and could be improved:r
    * Line 333: 'SAC is the most widely used RL technique for it strong performance...'
    * Line 376: 'SAC, which maximizes rewards through the interaction with the environment...'

**Questions:**

* The new successful experiences are incorporated into the buffer for both reinforcement learning and imitation learning, meaning these data are added to the buffer for Q-network updates twice. While these data might be good for behavior cloning, how might this affect the learning of accurate Q-functions?
* Why are the two tasks evaluated in separate simulators?
* What is the coefficient used to balance the objectives of SAC and Behavior Cloning?

---

### Note · Authors · 2024-11-27

**Comment:**

After careful consideration, we have decided to withdraw our submission. We believe that the focus of our work, which lies primarily in autonomous driving applications, aligns better with venues dedicated to such applications rather than conferences like ICLR, which emphasize advancements in artificial intelligence as a core discipline.

We are deeply grateful to the editors and reviewers for their valuable feedback and constructive comments, which will undoubtedly help improve the quality of our work as we prepare for future submissions.

**Withdrawal Confirmation:**

I have read and agree with the venue's withdrawal policy on behalf of myself and my co-authors.